# Does the State of Scientific Knowledge and Legal Regulations Sufficiently Protect the Environment of River Valleys?

Monika Konatowska [1], Adam Młynarczyk [2], Irmina Maciejewska-Rutkowska [1] and Paweł Rutkowski [1,*]

1   Department of Botany and Forest Habitats, Faculty of Forestry and Wood Technology, Poznań University of Life Sciences, Wojska Polskiego 71F, 60-625 Poznań, Poland; monika.konatowska@up.poznan.pl (M.K.); irmina.maciejewska@up.poznan.pl (I.M.-R.)
2   Environmental Remote Sensing and Soil Science Research Unit, Faculty of Geographic and Geological Sciences, Adam Mickiewicz University in Poznań, Wieniawskiego 1, 61-712 Poznań, Poland; adam.mlynarczyk@amu.edu.pl
*   Correspondence: pawel.rutkowski@up.poznan.pl; Tel.: +48-608295052

**Abstract:** The pressure of human activity in river valley environments has always been high. Even today, despite the increasing awareness of societies around the world regarding the need to protect water and biodiversity, there are concerns that the current river valley management systems are insufficient. Therefore, the aim of this study was to assess the state of knowledge about the soils and forest ecosystems of river valleys in terms of the possibility of protecting river valley environments. This study used data obtained from the Forest Data Bank (FDB) database, which focuses on forests in Poland. After analyzing 17,820 forest sections where the soils were described as fluvisols, it was found that forest areas associated with fluvisols (typical, fertile soils of river valleys) are quite well recognized and protected in Poland. Most (55%) forested fluvisols are located in Natura 2000 sites (an important European network of biodiversity hotspots), 4% in nature reserves, and 1% in national parks. Additionally, the main forest habitat type associated with fluvisols is riparian forest, composed mainly of *Quercus*, *Ulmus*, and *Fraxinus*, which is protected as Natura 2000 habitat type 91F0. Preserving the sustainability of the forest is also a form of soil protection. Despite the identification of soils and forests in river valleys, as well as appropriate legal tools, their protection may be ineffective due to the fragmentation of forms of protection and the lack of a coherent system for managing river valleys. Because the conservation status of the river valleys is also influenced by the management of areas located outside the river valleys, in order to protect river valley ecosystems, integrated conservation plans for entire catchments should be implemented. Due to potential conflicts related to the management of areas with diverse expectations of local communities, it would be advisable for such plans to be created by local experts but under the supervision of a specialist/specialists from outside the area covered by a given river basin.

**Keywords:** alluvial soils; management plans; protection river basins; Natura 2000 network

## 1. Introduction

Rivers are burdened by multiple stressors. According to the European Environment Agency [1], only about 40% of the regularly monitored European river water bodies were shown to have good ecological status in 2015. Intensive agriculture is one of the sources of these river stressors [2–6]. In turn, forests are indicated as a kind of land use that helps protect the quality of surface waters and the biota of river valleys.

Forests play an important role in the Earth's water cycle [7] and in soil protection. One of the important types of forest is riparian forest (RF) [8]. The importance of this forest ecosystem is emphasized by the inclusion of this habitat type in the European Natura 2000 network (nature protection areas in the territory of the European Union). The Natura 2000 network defines riparian forest natural habitats such as oak–elm–ash RF, ash–alder RF, and willow–poplar RF [9]. One of the main factors influencing the distribution and diversity of

riparian vegetation is soil [10–12]. The key type of soil in riparian areas is alluvial soil. In the World Reference Base for Soil Resources [13], these soils are classified as fluvisols (FLs), which, together with the soils of coastal and lakeside zones, cover over 5% of the total land area of the European Union [14].

The fertility of riverbeds depends on many factors, including the type of sediment, which is, in turn, related to the soil material in the catchment area, hydrological conditions, location along the river, distance from the river, or vegetation cover [15,16]. Generally, however, these soils are very fertile and, therefore, they often remain in agricultural use [17], which changes their structure and properties. Forest soils differ from agricultural soils in general due to a long period of undisturbed soil-forming processes, which is associated with the longevity of trees, including many generations of trees creating an uninterrupted forest environment. Therefore, RFs are an important source of knowledge on FLs; in turn, the knowledge about the transformation of alluvial soils has significant importance for agriculture and forestry [18].

Protection of FLs—the main soils of floodplains—is almost absent in research studies. According to El Hourani and Broll [19], only 44 published articles focus on soil protection in floodplains and riparian zones in North America, 25 in Europe, 12 in Asia, 12 in Oceania, 4 in South America, and 1 in Africa; however, among these 98 articles, most refer to physical and chemical properties of soil taxonomy, not of protection of these soils. These articles mostly concentrated on single countries (mainly the USA, Australia, Brazil, and China), and none of the articles reported data about soils in Poland. According to Schürings et al. [3], the territory of Poland, based on the background of other European countries, shows significant agricultural pressure. Therefore, it can be assumed that agriculture in Poland exerts equally significant pressure on river valleys.

The pressure of human activity in the river valley environment has always been high [20]. Even today, despite the increasing awareness of societies around the world regarding the need to protect water and biodiversity, there are concerns that the current river valley management systems are insufficient. There are also concerns regarding the protection of soil in river valleys [21]. The potential benefits arising from the soil–forest relationship in river valleys are probably underestimated due to the difficulty in estimating the natural function of the forest compared with its economic value. Therefore, the aim of this study was to assess, using the example of Poland, the state of knowledge about soils and forest ecosystems of river valleys in terms of the possibility of protecting the river valley environments.

This study considers the following questions:

- What is the share and distribution of forested fluvisols in Poland?
- What is the share of forested fluvisols in protected areas in Poland?
- What forms of protection are in place to protect forested fluvisols in Poland and is the protection sufficient?

The protection of forests (and soils associated with these forests) in river valleys varies from country to country and often depends on specific environmental regulations, conservation initiatives, and land management practices. Some general principles and methods of forest and river valley protection that are often applied in various countries around the world are as follows:

- Riparian buffer zones [22];
- Legal designations [23];
- Sustainable forest management [24];
- Biodiversity conservation;
- Community engagement [25];
- Watershed management [26];
- Conception of ecological corridors [27].

These strategies are further supported by some international conventions that allow for the protection of ecosystems in river valleys, such as the Water Framework Directive,

the Ramsar Convention, the Convention on Biological Diversity (CBD), the UN Framework Convention on Climate Change (UNFCCC), and the World Heritage Convention. These international agreements provide frameworks and mechanisms for countries to cooperate in the protection of river valley forests and the ecosystems they support, but specific strategies and regulations can vary widely depending on the ecological, cultural, and regulatory contexts of individual countries [28,29]. Therefore, theoretically, forests, and thus the soil under these forests, have a very strong legal base for their protection. In practice, these legal regulations are compounded by various social expectations and economic factors, which make the management of river valleys, and especially entire river catchments, one of the significant challenges of natural sciences in combination with economic and social aspects.

References to river valley management are more often found in strategic plans prepared for specific rivers, and less often in scientific publications.

Danci [30] showed the concept of alluvial forests with *Alnus glutinosa* and *Fraxinus excelsior* management in Maramureş Mountains Nature Park (Romania), but the management tools presented in the study demonstrate the conservation value of this habitat rather than a strategy of river basin management. An interesting topic of nature conservation, including RF, was presented by Pechanec et al. [31]. The authors highlighted an important question regarding conservation priorities: "conservation of species or natural processes?" This problem seems especially relevant to the 91F0 habitat [9], where oaks, elms, and ash trees are mentioned as the main forest tree species and where these species compete with each other and with other plants for light during the seedling development stage. Therefore, according to Palátová et al. [32], to achieve the success of oak regeneration on floodplains, it is necessary to remove the overstory immediately after the acorns fall. As a result, the most common method of *Quercus robur* regeneration on floodplains in the Czech Republic is artificial oak regeneration [33]. On the other hand, Marie-Pierre et al. [34] showed that the presence of *Fraxinus excelsior* in some mountain grasslands in the French Pyrenees depends on the management method. Cutting and a high intensity of grazing act negatively on natural ash regeneration, and a low grazing intensity acts positively.

Another important aspect of river valley management is the degree of fragmentation of forest complexes and their distance from each other, which, in terms of biodiversity protection, was pointed out by Filyushkina et al. [35] and Palmeirim [36]. The opinions are also divided in this case with regard to the question of whether a small number of large protected areas or a large number of smaller areas are more beneficial from the point of view of biodiversity conservation. Therefore, there are many reasons to investigate the environmental and economic aspects of conservation methods applied to riparian ecosystems. This study attempts to answer some of the current conservation questions in river valley management using the example of Poland.

## 2. Materials and Methods

### 2.1. The Distribution of Forested Fluvisols in Poland and Their Share in Various Forms of Nature Protection

A graphical illustration of the basic methodological data related to the distribution of forested fluvisols in Poland and their share in various forms of nature protection is presented in Figure 1.

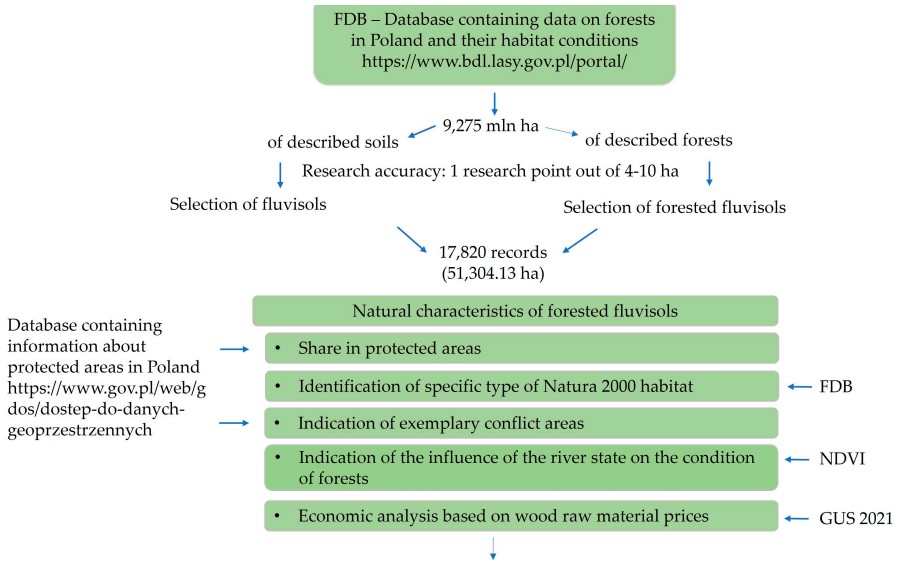

**Figure 1.** Flowchart of methods: FDB—Forest Data Bank [37]; NDVI—normalized difference vegetation index; GUS (2021)—Ref. [38].

The results of this study are based on data obtained from the Forest Data Bank (FDB), a database that collects data about forests in Poland. Every forest complex in Poland is inventoried every 10 years, with an accuracy of up to 1 ha, providing a basis for assessing changes taking place in nature and planning activities for the next decade. In our study, a total of 17,820 forest sections, where the soils were described as fluvisols, were analyzed. River valleys create an ecosystem with a high degree of complexity and relationships between biotic and abiotic factors, including both aquatic and terrestrial environments. In order to limit the number of variables taken into account in our study, the variables were limited to three factors: the type of soil, i.e., fluvisols, forest complexes occurring on fluvisols, and forms of protection in accordance with the Nature Protection Act in force in Poland [39] imposed on river valleys. These three variables were considered sufficient to demonstrate the degree of complexity of the river valley management problem.

However, the concept of fluvisol is not clear. In recent decades, the position of alluvial soils in Polish classification schemes has changed [18]. To ensure the comparability of results, soil research in all Polish forests has been carried out since 2000 in accordance with the "Classification of forest soils in Poland" [40]. In this classification, all soils with fluvic material were merged into the Fluvisols group, irrespective of the presence of other mineral diagnostic horizons or properties. All forest soils in Poland have been analyzed according to this classification, with an accuracy of one test point on at least 4 ha. The results of these studies have been entered into the Forest Data Bank [37].

All data were recorded in the form of attribute tables saved in ArcGIS 10.1 version as ".dbf file", in the layout shown in Table 1. ArcGIS is geographic information system (GIS) software that enables the storage, management, and retrieval of data.

**Table 1.** Part of the attribute table for fluvisol data.

| FID | Shape | Forest Address | Coordinates | | Area | Soil Subtype |
|---|---|---|---|---|---|---|
| | | | X | Y | [ha] | |
| 0 | Polygon | 02-11-3-12-240-o | 341,502 | 426,735 | 0.20 | Humic fluv. |
| 1 | Polygon | 04-12-2-14-148-a | 155,852 | 724,250 | 0.13 | Humic fluv. |

The attribute table presented in Table 1 contains the following columns:

- FID—a column automatically created by the program with the ordinal number of the object (Object ID); in ArcGIS, the first number is zero.
- Shape—a column contains information concerning the geometry type of the object, e.g., polygon, point, line, and others; this column is automatically created by the program.
- Forest address—a column created by users, this column presents the location according to the Polish system of administrative division of forests; the Polish system comprises the General Directorate supervising 17 Regional Directorates, which are divided into 432 Forest Districts. Forest Districts are divided into one, two, or sometimes three subdistricts. In each Forest District, the forest area in administrative and spatial terms is divided into numbered sections, usually rectangular, unless the terrain configuration requires a different division. In addition, the sections are divided into subsections marked with letters, covering homogeneous fragments of the forest. Section numbering always starts with 1 and subsections start with "a". According to this division, the forest address contains information about each unit, where the first example listed in Table 1 carries the following attribute definitions: 02—the Katowice Regional Directory; 11—the Kluczbork Forest District; 3—the Zameczek Subdistrict; 03—the Zawiść Forest Range; 240—the forest section number; "o"—the subsection. However, it should be emphasized that the published data contain forest addresses according to 2021, which may change in 10-year cycles based on the Polish system of forest state verification. As a result, the current forest address is 02-11-3-12-240-o.
- X and Y—the coordinates.
- Area (given in hectares).
- Soil subtype—this column contains soil data.
- Forest habitat type.

**Data Records**

All data were processed using ArcGIS v. 10.1 software and comprise 6 files with the following extensions:

- **.shp**—a mandatory Esri file that provides geometric feature data. Every shapefile has its own .shp file that represents spatial vector data, e.g., points, lines, and polygons in a map.
- **.shx**—a mandatory file used to search forward and backward.
- **.dbf**—a standard database file used to store attribute data and object IDs. A .dbf file is mandatory for shapefiles and can be opened using Microsoft Access or Excel [Microsoft®Excel®2021 MSO (Version 2403 Build 16.0.17425.20176) 64-bit was used].
- **.prj**—this file contains the metadata associated with the shapefile coordinate and projection system.
- **.sbn** and **.sbx**—these files optimize spatial queries and are saved together.

Finally, the table of attributes was exported to Excel, and the data informed the statements presented in this study (Tables 1 and 2).

**Table 2.** Share of forested fluvisols in total and in protected areas.

| Type of FLs | Share of Forested Fluvisols | | | | | | | |
|---|---|---|---|---|---|---|---|---|
| | Total | | In Natura 2000 Sites | | In National Parks | | In Nature Reserves | |
| | Area (ha) | % of Total Area | Area (ha) | % of Total RF or MF Area | Area (ha) | % of Total RF or MF Area | Area (ha) | % of Total RF/MF Area |
| River FLs (RF) | 51,302.77 | 100.00 | 27,857.07 | 54.30 | 681.86 | 1.33 | 2206.21 | 4.30 |
| Marine FL (MF) | 1.36 | 0.00 | 1.36 | 100.00 | 0.00 | 0.00 | 0.00 | 0.00 |
| Total | 51,304.13 | 100.00 | 27,858.43 | 54.30 | 681.86 | 1.33 | 2206.21 | 4.30 |

*2.2. Issues in the Protection of Riparian Forests Related to Fluvisols*

A key element of this work is the assessment of the conservation status of river valleys and forest-covered fluvisols. This problem is presented in Section 3.2, using the example of two key rivers in Poland, the Odra and the Warta. The conservation status assessment example considers the methods of protecting selected parts of river valleys and adjacent areas, as well as the threats affecting the functioning of rivers.

Data on the boundaries of protected areas in Poland (national parks, nature reserves, Natura 2000 areas, and wildlife corridors) were downloaded from the public website "https://www.gov.pl/web/gdos/dostep-do-danych-geoprzestrzennych" (accessed on 17 April 2024) managed by the General Directorate for Environmental Protection, which is the government agency (gov.) responsible for environmental protection in Poland.

The relationships between the NDVI index, vegetation cover, and the moisture of forest habitats used to the data analysis methodology are described in more detail in the article published by Młynarczyk et al. [41].

Data used to demonstrate the relationship between the condition of forest stands and the water level in the Odra River were retrieved from https://danepubliczne.imgw.pl/data/dane_pomiarowo_obserwacyjne/dane_hydrologiczne/dobowe/ (accessed on 17 April 2024). Daily data were used, but the folder structure did not allow for automatic utilization. The daily data were loaded into RStudio and the Nietków hydrological station was filtered out. In the next step, the data were merged into a single series and loaded into Microsoft Excel (Version 2403 Build 16.0.17425.20176) 64-bit for visualization.

## 3. Results

*3.1. The Distribution of Forested Fluvisols in Poland and Their Share in Various Forms of Nature Protection*

Forested fluvisols in Poland consist almost exclusively of soils composed of river sediments (Table 1). Most (54.3%) fluvisols occur at the Natura 2000 sites. Of these soils, 1.33% are noted in national parks, and 4.30% are noted in nature reserves (Table 2). Their distribution is shown in Figure 2.

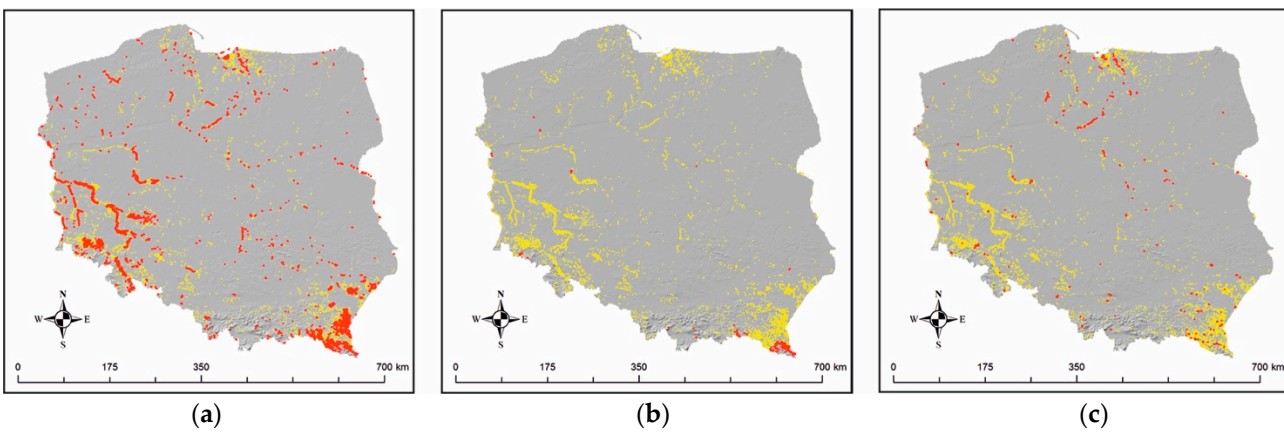

|           (a)           |           (b)           |           (c)           |

**Figure 2.** Distribution of forested FLs: (**a**) Natura 2000 sites; (**b**) national parks; (**c**) nature reserves. Red—FLs in the given protected area type; yellow—FLs out of the given protected area type.

Types of natural habitats are an important element of the Natura 2000 network. Of the many natural habitat types mentioned [9] in the river valleys of Central Europe, two play a key role: the oak–elm–ash forest, code 91F0; the ash–alder forest, code 91E0.

The share of RFs in various forms of nature protection and divided into 91F0 and 91E0 habitats is presented in Table 3.

**Table 3.** Share of forested FLs related to 91F0 and 91E0 Natura 2000 habitat types: in total and in protected areas.

| Natura 2000 Forest Habitat Type | Share of Natura 2000 Habitat Type (%) | | | |
|---|---|---|---|---|
| | Total | In Natura 2000 Areas | In National Parks | In Nature Reserves |
| 91F0 (the oak–elm–ash forest) | 75 | 78 | 94 | 86 |
| 91E0 (the ash–alder forest) | 3 | 3 | 0 | 2 |
| Other (non-Natura 2000) habitat types | 22 | 19 | 6 | 12 |
| Total | 100 | 100 | 100 | 100 |

It should be noted that the given "zero" (in fact 0.08%) share of habitat type 91E0 in national parks shown in Table 3 does not mean that there is no habitat type 91E0 in national parks in Poland; rather, this means that this habitat type does not occur in FLs in national parks. It can be assumed that this corresponds to the ecological role of FLs and their strong correlation with habitat type 91F0. Habitats of type 91E0 are generally found on other soil types. Therefore, it can be assumed that fluvisols form actual or potential habitat type 91F0 and, vice versa, that a typical soil for habitat type 91F0 is FL.

Assuming that national parks in Poland have the best status in terms of identifying abiotic and biotic components of nature, the relationship of FL–RF in national parks can be taken as a model state of 91F0 habitat, with all natural processes that occur in this type of forest. The same national parks play a special role in the protection of RF.

The conservation management strategy of FLs and RFs depends on their location within or outside of existing conservation forms in Poland. Approximately 5% of forests in FLs are protected in Polish national parks and nature reserves (Figures 2 and 3). The role of national parks and nature reserves presented in Figure 3 is obvious; they must protect the best-preserved components of nature and ensure the natural course of processes taking place in nature. However, the primary areas of RF and FL protection are Natura 2000 sites, where approx. 55% of all FLs associated with forests in Poland occur. As shown in Table 2, the main type of natural habitat on FLs is 91F0, which is composed mainly of oaks, elms, and ash trees. Forest management in Poland in relation to RFs includes activities similar to the methods of active protection of habitats 91E0 and 91F0. Therefore, in Figure 3, they are combined into one unit of benefits and losses, although, of course, an important difference in the conditions of forest management is timber harvesting. The value of alluvial forest habitats is estimated in Table 4.

**Table 4.** The monetary value of alluvial forest habitats in protected areas in Poland.

| Economic Data | Forest Alluvial Habitats | | |
|---|---|---|---|
| | In Natura 2000 Sites | In Nature Reserves | In National Parks |
| Area (ha) | 27,858.43 | 2206.21 | 681.86 |
| Average wood prices in Poland per 1 m$^3$ in EUR (in PLN by GUS 2021 [38], calculated to EUR according to the European Central Bank [42]). | | 44.30 | |
| Average resources of gross timber (in m$^3$ per 1 ha) of broadleaved tree species managed by the State Forests in Poland [38]. | | 247 | |
| Economic forest value in EUR. | 304,829,727 | 24,140,570 | 7,460,980 |

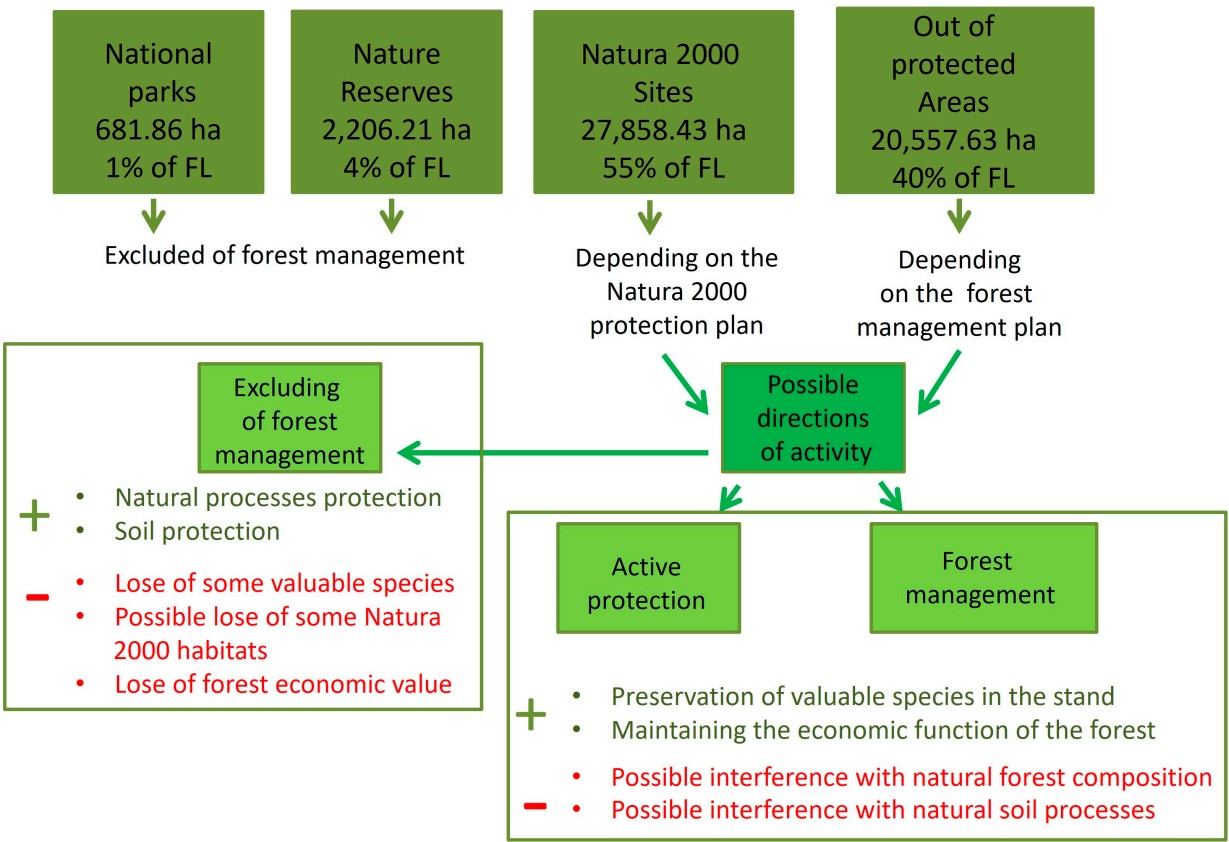

**Figure 3.** Management scheme for the protection of river valleys.

The economic losses shown in Table 4, which are relatively easy to calculate on the basis of the value of wood and the prices of different wood assortments, are difficult to compare with the natural value of forests. Smith and Smith [43] list eight RF functions, each of which are difficult to directly estimate economically:

1. Protecting water quality by trapping sediment;
2. Protecting water quality by reducing nutrient and pollutant inputs;
3. Reducing stream bank erosion;
4. Providing shade and modifying stream temperatures;
5. Providing organic matter for aquatic ecosystems;
6. Enhancing local biodiversity;
7. Providing wildlife corridors;
8. Enhancing aesthetic, recreational, educational, and scientific values.

It is possible to indicate the share of FLs in ecological corridors (Figure 4), where they occupy 31,078.91 ha (60.6% of the total forested FL area in Poland).

As seen from the map shown in Figure 4, a significant part of forested FLs is outside wildlife corridors. It should also be emphasized that the wildlife corridors shown in Figure 4 are not limited to only river valleys but also include forest complexes outside floodplains; therefore, it is natural that FLs do not exist outside of terrains flooded by rivers.

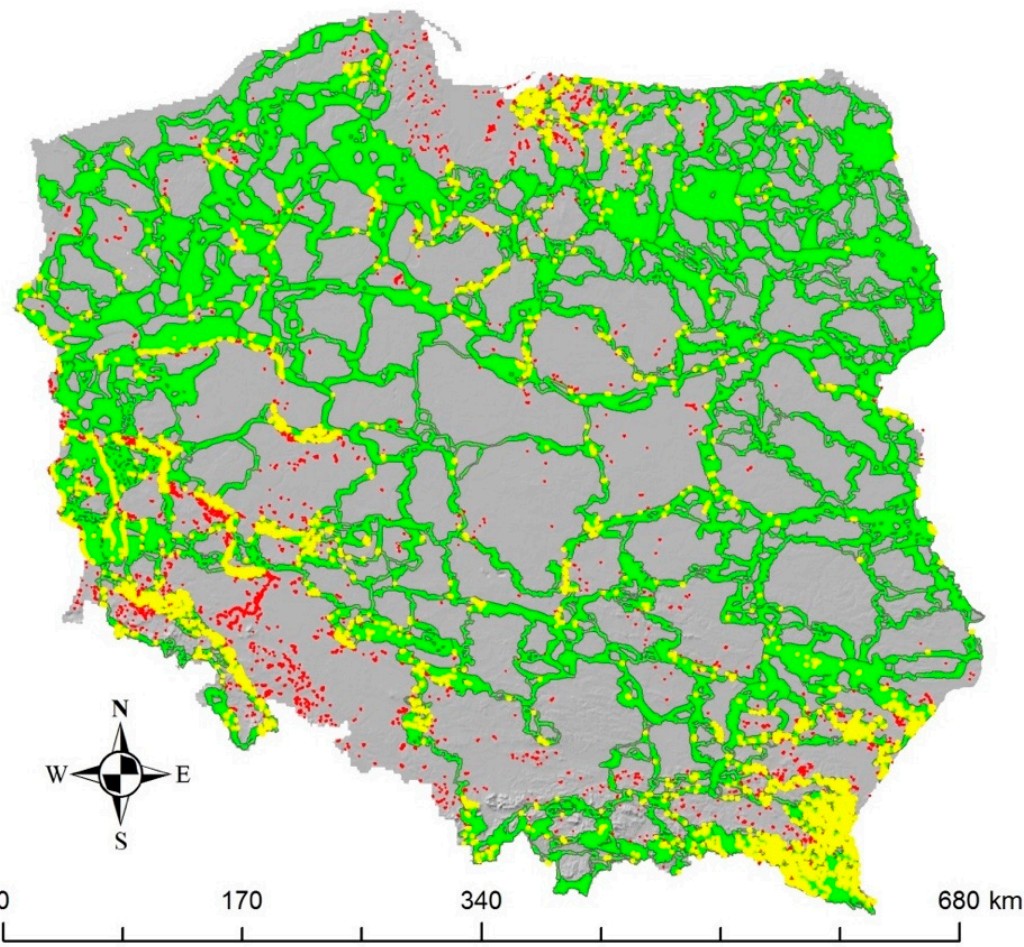

**Figure 4.** Distribution of forested FLs on the background of wildlife corridors (green); red—FLs out of wildlife corridors; yellow—FLs in wildlife corridors (source of map of wildlife corridors: https://www.gov.pl/web/gdos/dostep-do-danych-geoprzestrzennych) (accessed on 17 April 2024).

### 3.2. Issues in the Protection of Riparian Forests Related to Fluvisols

The enhancement of local biodiversity as an important function of RFs as mentioned by Smith and Smith [43] is beyond a doubt. However, a problem arises when we are dealing with various forms of nature protection and various nature protection goals. Rivers and their environments form ecosystems that are particularly difficult to manage. On the Polish scale, an example may be the Warta River, along which there are national parks, nature reserves, Natura 2000 habitat sites, and bird Natura 2000 sites (Figure 5). In addition, a number of rivers are of a transboundary nature, such as the Odra River shown in Figure 5, which requires additional regulations in the field of environmental protection, including international ones. Near the mouth of the Warta River to the Odra River, the highest forms of nature protection in Poland were created—the Ujście Warty National Park and the Natura 2000 PLC080001 site, which were designated to ensure the combined protection of habitats and birds (Figure 5).

The fluctuating water level in rivers, which may depend on both natural and anthropogenic causes, affects the condition of the stands, which is illustrated in Figure 6 using the eastern part of the PLH080028 Natura 2000 site (also marked in Figure 5) located along the Odra River as an example. In Figure 6, the condition of stands depending on the fluctuating water level was assessed on the basis of the NDVI index (normalized difference vegetation index).

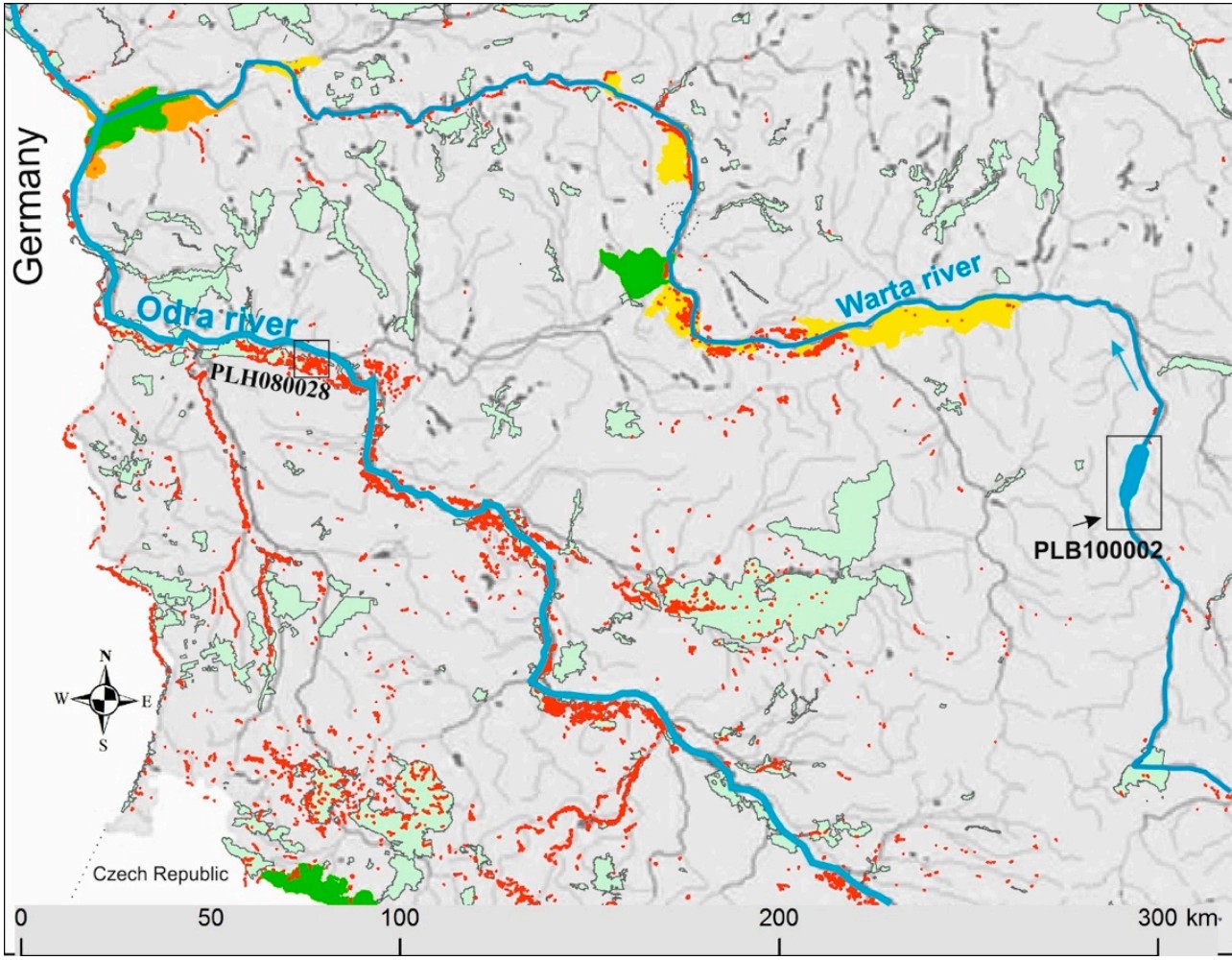

**Figure 5.** Potential conflict in the protection of riverside areas in the example area of the Warta River—Natura 2000 site PLB10002 (box); red—forested fluvisols; blue—rivers; yellow—Natura 2000 PLH sites along the Warta River; orange—Natura 2000 site PLC080001; dark green—national parks; light green—other PLH Natura 2000 sites. The Eastern part of the PLH080028 Natura 2000 site is indicated by a black square (referred to in Figure 6).

Figure 6a (as of 3 July 2018) and Figure 6b (as of 3 June 2019) show the condition of the stand at a water level of 200 cm. Figure 6c (as of 7 August 2018) and Figure 6d (as of 29 August 2019) illustrate the condition of the stand at the water level of approx. 150 cm. Figure 6e (as of 3 September 2018) and Figure 6f (as of 1 September 2019) show the condition of the stand at the river level within 110 cm. Any changes in the flow level in rivers, which due to human activity may overlap with natural factors, may affect the condition of stands. For stands along the Odra River, in the presented area, the critical water level in the river can be assumed to be 100 cm.

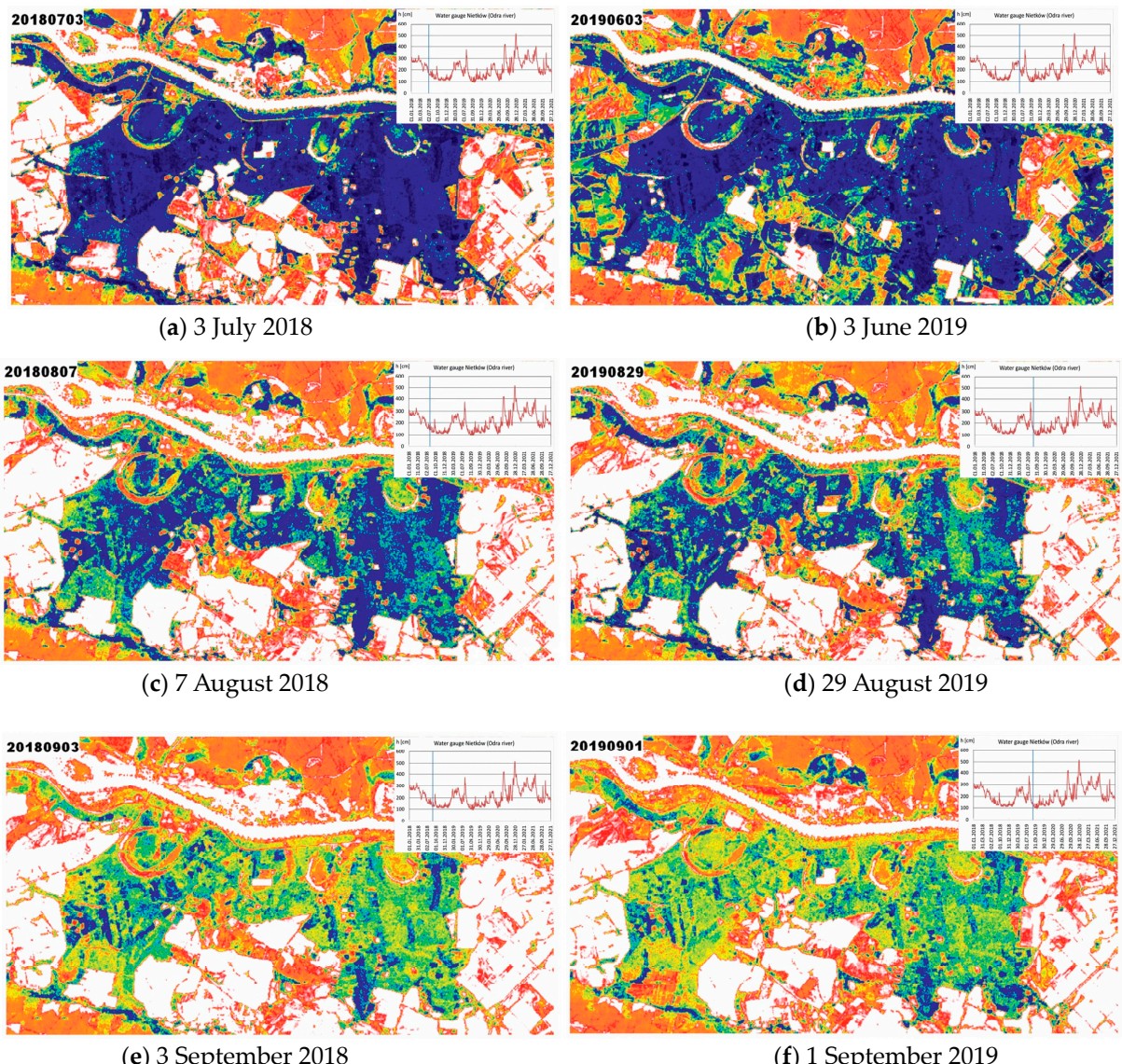

**Figure 6.** Changes in the condition of stands in riparian forests in the eastern part of the PLH080028 Natura 2000 site in 2018 and 2019 depending on the level of the Odra River. In the upper right corner of each subfigure, the changes in the water level in the river are marked, and the vertical blue line is the date for which each of the subfigures was performed. Figures (**a**–**f**): dark blue color—broadleaved stands in good condition; yellow—broadleaved stands in bad condition.; orange—drier pine forests or non-forest vegetation in overdried habitats.

Among the RF functions mentioned by Smith and Smith [43], one should also pay attention to "reducing stream bank erosion". In the case of riparian forests occurring in Poland, this is of particular importance as most fluvisols covered with forests are located in mountain areas, in the upper sections of rivers, or in hill regions of the lowland part of the country (Figure 7), where the problem of erosion is extremely important.

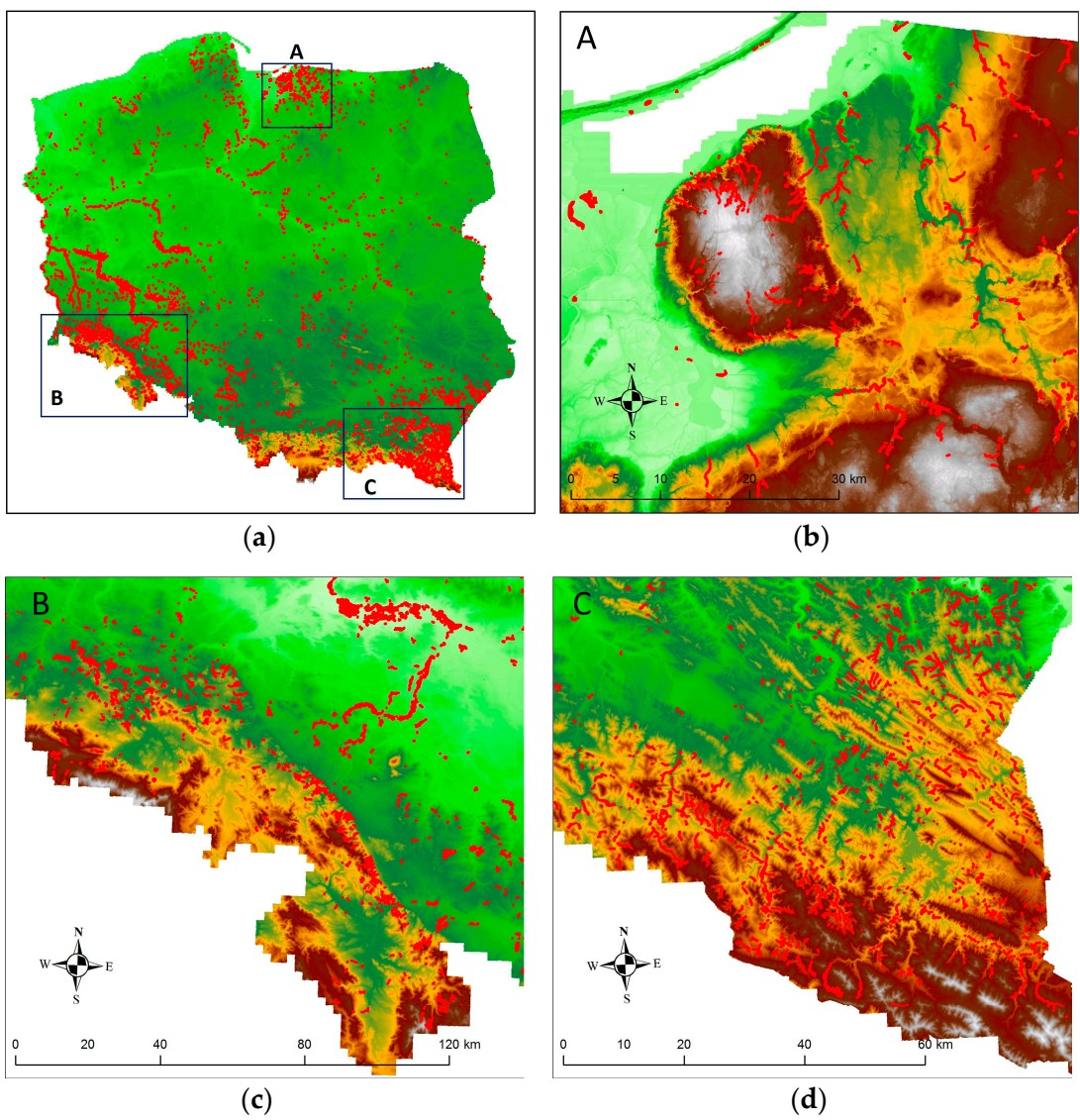

**Figure 7.** Forested fluvisols (red areas) on the background of a relief map of Poland in the system of dynamic hypsometry: (**a**) general map of forested fluvisols in Poland; (**b**) Wysoczyzna Elbląska region (A on (**a**)); (**c**) Sudety mountain region (B on (**a**)); and (**d**) Bieszczady mountain region (C on (**a**)) In the Figure A, white color indicates the highest elevation of the terrain, brown indicates height elevation, and green indicates the lowest elevation; in Figures B and C brown color indicates the highest elevation, green the lowest.

## 4. Discussion

Riparian forests in Poland in relation to the background of the worldwide RF and fluvisol network. FLs occupy less than 350 million ha worldwide, mainly along great rivers, such as the Amazon, Ganges, or Nile [13]. In Europe, this type of soil covers over 5% of the total land area of the European Union [14]. Therefore, Poland is only a small part of the world FL and RF network (51,304.13 ha; 0.55% of the total forest area in Poland), but the management of forest habitats related to FLs is important in the European system of nature protection. In particular, the environmental value of alluvial forests is much higher than that of their area. Rivers and river valleys, apart from many other functions [43], play the role of ecological corridors. They are also an important natural link between Natura 2000 sites and, as shown in the research results, the largest share of FLs in Poland is in Natura 2000 sites (55% of FLs in Poland). It is also worth paying attention to the disproportion between the occurrence of FLs and RFs. As previously reported, FLs cover over 5% of

the total land area of the European Union [14] (according to Clerici et al. [44], RF covers approximately 2% of the European continental area; the same share is given to the Nordic countries [45]). Therefore, the area of FLs is two and a half times higher than that of riparian forests, which can be explained by the importance of FLs in agriculture. These soils are generally fertile, and therefore, they often remain in agricultural use. This comparison shows that the protection of forests in FLs is even more important because, among many other functions, forests protect the features specific to FLs.

Additionally, the status of marine fluvisols should also be clarified. The forested form of marine fluvisol, provided in Table 1, has been recorded in one place in Poland only, at the mouth of Błądzikowski Creek (in Polish: Potok Błądzikowski) to Pucka Bay (in Polish: Zatoka Pucka). The affiliation of this position with marine FLs is debatable. As reported by Olszak and Jereczek-Korzeniewska [46], the mouth of the stream is an alluvial fan formed as a trace of water outflow from the melting ice sheet. Therefore, one marine FL habitat in Poland requires more detailed research.

The dominant share of forested fluvisols (and not the fluvisols themselves) in Natura 2000 areas results from the criteria for diagnosing habitat type 91F0 (riparian mixed forests of *Quercus robur*, *Ulmus laevis* and *U. minor*, and *Fraxinus excelsior* or *F. angustifolia* along the great rivers), which is one of the types mentioned in the Habitats Directive [47] and is identified as one of the two pillars of the Natura 2000 network. According to the characteristics contained in the Interpretation Manual of European Union Habitats [9], habitat type 91F0 is "Forests of hardwood trees of the major part of the river bed, liable to flooding during regular rising of water level or, of low areas liable to flooding following the raising of the water table. These forests develop on recent alluvial deposits". Therefore, alluvial forests are naturally included in the area of interest related to the Natura 2000 network. In addition, river valleys create areas where a number of other valuable natural habitats can be found, such as alluvial forests with *Alnus glutinosa* and *Fraxinus excelsior* (code 91E0), or rivers with muddy banks with *Chenopodion* and *Bidention* vegetation (code 3270) [9]. Combined with the natural value of the river itself, this means that river valleys are often protected under the Natura 2000 network by designating Natura 2000 areas along them. Therefore, because habitat type 91F0 is associated with riverine forests and riverine forests are associated with fluvisols, hence the significant share of habitat type 91F0 and fluvisols in Natura 2000 areas presented in our study.

Environmental and monetary value of alluvial forest habitats. There are very few publications on the natural value of FLs. According to El Hourani and Broll [19], in a review of the literature on these soils, most of the works refer to their physical and chemical properties or systematics. However, the environmental value of RFs is much more clearly reflected in the literature [35,48–52]. Summarizing the natural value of river valleys is difficult because a number of studies are based mainly on single or several individual rivers [35]. In Poland, authors described RFs in the Sudety Mountains, at the border between Poland, the Czech Republic, and Germany. As our research shows, this is one of the main RF regions in Poland (Figure 6c). According to Perzanowska and Korzeniak [50], this region is the most species-rich at both local and regional scales. In addition, our study shows that forests in FLs are mainly associated with the natural habitat type 91F0 in Poland. According to the Interpretation Manual of European Union Habitats [9], this type of forest habitat occupies basins along the great rivers. Most FLs noted in the Polish Forest Data Bank are connected with mountain creeks and small rivers. Our research has shown that the second natural habitat type associated with FLs in Poland is 91E0. Both natural habitats (type 91F0 and 91E0) are listed in the Red List of Natura 2000 habitat types of Poland [50] and designated in the "Vulnerable" group.

An interesting attempt to calculate the natural value of natural forest habitats in protected areas is presented in a scientific article published by Pechanec et al. [52]. The authors estimated, among others, the monetary value of hardwood forests of lowland rivers. The value of this habitat type, reported in Special Areas of Conservation in the Czech Republic as occupying an area of 117.72 km$^2$, was estimated at about EUR 4.5 billion. According to

our research, assuming that the area of this type of forest in Poland is 278.58 km$^2$ (Table 3), the monetary value would be almost EUR 11 billion. This value is over 35 times higher than the value of wood estimated in Table 3 (EUR 304,829,727) that could be obtained from Polish forests growing on fluvisols. Therefore, even if both calculations may be debatable, they still show that the natural monetary value of forest habitats exceeds the value of wood.

Selection of regions and goals of nature protection in river basins. Because of limited labor forces and funding, it is not possible to protect all land areas; therefore, the problem of how to select essential regions and protect biodiversity is discussed [53]. One of the important questions highlighted by Filyushkina et al. [35] is as follows: what is better, a few large or many small conservation areas? There is no simple answer to this question because river valleys are incomparable at the world scale. It is difficult to compare the Yangtze River Basin (1.8 million km$^2$) [53] with Poland as the entire area of the country is just over 300,000 km$^2$.

Pechanec et al. [31] focused on another important topic of nature protection: the fragmentation of protected areas. According to their study, nearly all protected areas in the Czech Republic are smaller than 10 km$^2$. In our study, we identified 17,820 forested FLs. This means that the average area of one riparian forest habitat is less than 3 ha (the total area of all investigated forests is 51,304.13 ha). This is due to the natural arrangement of forests in the narrow river valleys in Poland, but it is not conducive to effective protection of the natural processes taking place in river valleys. Despite all the presented discrepancies, in Poland and in other European countries, the selection of regions is more or less regulated by a network of national parks, nature reserves, Natura 2000 sites, and other forms of protection. However, it is more difficult to find a consensus on the purpose of protecting a given protected area and protecting river basins along the entire river course. An example may be the divergent goals in the Warta River valley, where along the same river there is a dam reservoir constituting the Natura 2000 site designated for birds, national parks protecting the broad natural values of the region, and other protected areas designated for the protection of river valleys and their natural habitats (Figure 5). The Natura 2000 PLB 100002 site (the Jeziorsko Reservoir), created on an artificial water reservoir formed as a result of damming the river, acts on the entire Warta River basin. The natural flow of the river was disturbed by the damming of water in the retention reservoir, which affects the floodplains located further down the river [54]. Such cases should require an integrated management plan for natural assets along the entire river course and not separate plans for each area. The impact of the Jeziorsko Reservoir is mentioned in the draft of the Ujście Warty National Park protection plan [55] as one of the existing external threats. In protective tasks for the "Ujście Warty" National Park [56], this threat has been removed for unknown reasons, but it undoubtedly matters. However, there is no comprehensive assessment of the impact of the Jeziorsko Reservoir on the RFs along the entire course of the river, along which, as a result of the research, a total of 1338 RF complexes on the FLs were recorded.

Management of the river valleys. As stated earlier [14,44], the data on the share of FLs in relation to riparian forests in Europe (5% vs. 2%) show that in river valleys, forests are not the dominant form of landscape, generally giving way to agricultural land. Our research shows that forested FL cover in Poland is only 0.55% of the total forest area, 60% of which is already in the areas with the highest protection status (national parks, nature reserves, and Natura 2000 sites) (Figure 3). In national parks and nature reserves, these forests are excluded from use (Figure 3). However, it is debatable whether this is to be a form of strict protection or a form of active protection in response to current threats. It is well known that rivers are the highways to invasive species and, in many cases, the control and removal of invasive plants are necessary [57]. An overview of invasive alien plants in Poland's national parks was presented by Bomanowska et al. [58]. Therefore, the question of strict protection of RFs (and not only riparian forests) may be an extension of the question of Pechanec et al. [31] on conservation priorities. It is worth adding that the term strict protection is interpreted differently but generally means the exclusion of human activities. According to [59], strict protection "means ensuring minimal human intervention in order

to allow the natural forces and processes to predominate". In Poland, strict protection means the total and permanent abandonment of direct human intervention in the state of ecosystems, in the formations and components of nature, and in the course of natural processes in protected areas. Additionally, in the case of species, strict protection refers to the year-round protection of individual species and their stages of development [39]. Therefore, in the case of ecosystems with disturbed ecological balance, the question can be raised of whether strict protection, meaning long-lasting and undisturbed protection of natural processes, is truly the best form of protection.

In view of the above threats to riparian forests, the scope of their forest management seems to be one of the less important factors affecting the condition of forest ecosystems along rivers, provided that the continuity of the forest is maintained and, if possible, the soil cover is untouched. The rules of forest management in Poland in relation to riparian forests generally maintain these conditions. However, the method of forest management in riparian forests, which differs depending on the target species composition of the stand, may be discussed. If the aim is to protect a spontaneously formed species composition, then such forests can be excluded from forest management. If the aim is to focus on oak or ash, actions supporting both tree species are required. In the case of spontaneous, natural processes taking place in the forest, oak often loses competition with elms, ash, and other species at the stage of natural regeneration [32]. Ash in habitat type 91E0 dies due to the pressure of animals, as well as due to the expansion of ash dieback [60]. Both oak and ash trees can be maintained by carrying out an active form of conservation through protective measures, including protection against herbivores and protection of natural regeneration against competition from other plant species. If the 91F0 habitat were to be excluded from any human interference, the share of pedunculate oak in 91F0 would be expected to decrease. The presence of *Quercus robur* in the stand is not only a source of valuable wood material but also, and perhaps above all, a habitat for species valuable from the natural point of view, such as *Osmoderma eremita* (in Natura 2000 code 1084) or *Lucanus cervus* (code 1083) [61]. The sum of all these issues makes the protection of FLs and riparian forests a complex decision-making process, encompassing environmental, social, and economic issues, which basically fully corresponds to the preamble to the Habitats Directive [47], which states "... the main aim (...) of this directive being to promote the maintenance of biodiversity, taking account of economic, social, cultural and regional requirements". Referring to all the abovementioned issues, it can be concluded that riparian forests and related fluvisols formally have a sufficient legal basis for their protection, although the effectiveness of this protection may be debated. Considering the various protection objectives, active forms of this protection should not be excluded from national parks and nature reserves. In Natura 2000 sites and forests that are not covered by any form of protection, forest management should also be acceptable. However, forest management should consider the obligation to maintain the continuity of the forest and care for the maintenance of intact soil cover. The only exception to the obligation to maintain forest continuity could be natural factors, such as hurricanes, fires, or extreme floods.

Considering the results of the research showing that the largest share of forested FLs in Poland is in Natura 2000 sites, it seems that none of the forest ecosystems in Poland fit into the idea of the Natura 2000 network as strongly as the riparian forests.

Challenges and barriers faced in implementing integrated conservation plans for FLs in river basins. At least in Poland, there are no legal, administrative, or institutional barriers that would limit the possibility of creating integrated protection plans for entire catchments. Moreover, there are management plans for catchments of large rivers, but they concern flood risk management plans, focusing less on the protection of the natural values of the river and its catchment [62]. Attempts are being made to create protection plans for entire catchments, but they concern small rivers [63]. Attempts are being made to develop protection plans, but even in the case of short streams (44 km), they focus on the river itself and, to a lesser extent, on its catchment area [63]. It seems that the limitations here are the human resources necessary to describe large areas of the catchment, requiring a wide

team of specialists, as well as the necessary time and costs associated with this type of study. These limitations may be gradually minimalized in the future owing to technological progress and increasingly better nature databases. However, it seems that the basic problem of managing large river catchments is the divergent interests of various social groups, which are difficult to reconcile. As shown in the results (Section 3.2), even in the case of a relatively small river such as the Warta (about 800 km long), there are natural conflicts between various forms of nature protection, and these conflicts do not yet refer to various forms of land use (mining industry, agriculture, urbanization, and a number of others). It would also be necessary to take into account different social expectations toward different managers supervising large areas, such as river catchments. Perhaps the solution to this problem would be to develop river catchment management plans produced by local experts, but under the supervision of specialists not associated with a given region or even a country. However, in such plans, riparian forest–soil relationships must be taken into account, as indicated by the results of Celentano's research [64] confirming that the degradation of riparian forests has a severe negative effect on soil properties and subsequent ecosystem services. This also applies to the replacement of forests with agricultural use, which also affects the soil macrofauna in fluvisols [65].

## 5. Conclusions

In this study, the issue of managing river catchments was investigated in the example of Poland, a country with a relatively high level of knowledge about the environment, its natural values, and its threats, where every forest complex is inventoried every 10 years, with an accuracy of up to 1 ha, providing a basis for assessing changes taking place in nature and planning activities for the next decade. The country also has an appropriate legal system that theoretically protects the natural environment against degradation. Despite these objectives, even the location of riparian forests in protected areas does not guarantee their proper protection. Taking into account our results, the following conclusions can be drawn from our study:

- Riparian forests have high natural value, protecting both biodiversity and natural processes occurring in the associated soils. Therefore, whenever possible, river valley management plans should take into account the preference for this form of land use over others, especially since, as shown in our study, the natural value of riparian forests may significantly exceed their economic value. Moreover, in river valley management plans, it would also be worth taking into account the value of riparian forests as natural flood areas, which are intended to absorb water during periods of river flooding, and including this value in the economic balance related to the management of river valleys. It would also be worth taking into account the role of forests in protecting natural processes occurring in soils.
- Our study shows that riparian forests dominate in mountainous areas in Poland, which is probably due to the poor availability of the mountain regions for agriculture. However, this emphasizes the role of forests on fluvisols in counteracting erosion.
- Theoretically, forest areas associated with fluvisols (typical, fertile soils of river valleys) are quite well recognized and protected in Poland. Most (55%) forested fluvisols are located in Natura 2000 sites (an important European network of biodiversity hotspots), with 4% in nature reserves, and 1% in national parks. Additionally, the main forest habitat type associated with fluvisols is riparian forest, which is composed mainly of *Quercus*, *Ulmus*, and *Fraxinus*, and is protected as Natura 2000 habitat type 91F0. Preserving the sustainability of the forest is also a form of soil protection.
- Despite the identification of soils and forests in river valleys, as well as appropriate legal tools, their protection may be ineffective due to the fragmentation of forms of protection and the lack of a coherent system for managing river valleys. The lack of such plans results in a lack of control over the condition of rivers and divergent goals of activities in the field of individual objects of protection occurring in river valleys (fauna, flora, natural habitats, and soils).

- Because the conservation status of the river as well as forest and non-forest ecosystems associated with river valleys is also influenced by the management of areas located outside the river valleys, integrated conservation plans for entire catchments, not just river valleys, should be implemented in order to protect river valley ecosystems. Due to potential conflicts related to the management of areas with diverse expectations of local communities, it would be advisable for such plans to be created by local experts, but under the supervision of a specialist/specialists from outside the area covered by a given river basin.
- In the case of large rivers, limited resources of expert staff, high development costs, and limited time allocated for this type of development may be a barrier to developing management plans for entire catchments. However, it is assumed that significant technological progress facilitates the monitoring of land and water areas, as shown, among others, in the example of using the NDVI index to assess the relationship between the state of the river and the condition of riverside forests.

**Author Contributions:** M.K. and A.M. conceived the ideas and designed the methodology. All authors conducted the work and prepared the manuscript. All authors have read and agreed to the published version of the manuscript.

**Funding:** This research received no funding.

**Data Availability Statement:** The raw data supporting the conclusions of this article will be made available by the authors on request.

**Acknowledgments:** The authors gratefully acknowledge the Office of Forest Management and Geodesy (Biuro Urządzania Lasu i Geodezji Leśnej) and the General Director of State Forests for providing data from the Forest Data Bank (Bank Danych o Lasach).

**Conflicts of Interest:** The authors declare no conflicts of interest.

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
