# Peer review of "Does the State of Scientific Knowledge and Legal Regulations Sufficiently Protect the Environment of River Valleys?"

_land, doi:10.3390/land13050584_

Round 1
Reviewer 1 Report (Previous Reviewer 1)
Comments and Suggestions for Authors
This manuscript has been resubmitted and revised according to the reviewer's suggestions. In my opinion, the paper has been improved and most of my concerns about the previous version have been addressed. I therefore recommend it for publication in Land.
Author Response
Dear Reviewer
Thank you very much for recommendation of our manuscript for publication in Land.
Best regards
Authors
Reviewer 2 Report (Previous Reviewer 2)
Comments and Suggestions for Authors
The authors have improved the original manuscript by changing the focus of their research. The addition of research questions helps the readers to find out the purpose and originality of the proposal. However, some aspects should be corrected, modified or explained before considering it for publication.
Title: It is a good title, but should be more specific (Where?)
Introduction: It is straight-foward. I suggest finishing this section with the objectives of the research.
Material and Methods: The authors have used different geospatial databases in a GIS environment, however they should add more information about these databases. Also the validation using ground truth and the statistical analysis is missing.
Conclusions: They should be more concise and related to the research questions. The first one is a supposition (Natural value>Economic value). The second one (location of the riparian forest) is irrelevant. What is the evidence for the fourth conclusion (fragmentation forms)
In summary, which is the answer to the question: Do the state of scientific knowledge and legal regulations sufficiently protect the environment of river valleys?
Author Response
We would like to thank a lot the Reviewer for the review. Below are our responses to the Reviewer's comments given in the second round of the revision.
The authors have improved the original manuscript by changing the focus of their research. The addition of research questions helps the readers to find out the purpose and originality of the proposal. However, some aspects should be corrected, modified or explained before considering it for publication.
Reviewer's comments: Title: It is a good title, but should be more specific (Where?)
Authors' response: We believe the problem affects all rivers, so we left the title unchanged.
Introduction: It is straight-foward. I suggest finishing this section with the objectives of the research.
Authors' response: We think the section following after objectives is also important, so we left it. But if the Editor decides that, according to the Reviewer's suggestion, the section after objectives should be removed, we will remove it.
Material and Methods: The authors have used different geospatial databases in a GIS environment, however they should add more information about these databases. Also the validation using ground truth and the statistical analysis is missing.
Authors' response: We added more information about the databases
Conclusions: They should be more concise and related to the research questions. The first one is a supposition (Natural value>Economic value). The second one (location of the riparian forest) is irrelevant. What is the evidence for the fourth conclusion (fragmentation forms)
Authors' response: The statement "the natural value of riparian forests may significantly exceed their economic value" is written in supposition mode, but we have explained it in the Discussion section, in the subsection "Environmental and monetary value of alluvial forest habitats", based on the results of research by Pechanec et al. [47 ] We used the supposition mode assuming that our statement is true if the data of Pechanec at al. are correct.
According to the question: "What is the evidence for the fourth conclusion (fragmentation forms)" we believe that we have shown it in Fig. 5 by assuming that if there are many forms of protection along one river (national park, Natura 2000 areas, nature reserves) and each of these forms has a separate protection plan, then - in our opinion - this is a fragmentation of forms of protection
In summary, which is the answer to the question: Do the state of scientific knowledge and legal regulations sufficiently protect the environment of river valleys?
Authors' response: In our opinion the answer is on the conclusion: “Despite the good identification of soils and forests in river valleys, as well as appropriate legal tools, their protection may be ineffective due to the fragmentation of forms of protection and the lack of a coherent system for managing river valleys”
Reviewer 3 Report (Previous Reviewer 3)
Comments and Suggestions for Authors
Thanks for addressing all my comments and questions.
Comments on the Quality of English LanguageFair enoughg
Author Response
Dear Reviewer
Thank you very much for your efforts in reviewing of our paper. Before the final version of the manuscript will be submitted, the text will be checked by native speaker.
Best regards
Authors
Reviewer 4 Report (New Reviewer)
Comments and Suggestions for Authors
General
The manuscript titled “Do the state of scientific knowledge and legal regulations sufficiently protect the environment of river valleys?” is a study about Polish riparian forest habitats and about their conservation issues. It is important to draw attention to the protection of these ecosystems but in a better way than in this manuscript. This manuscript is not so similar to a scientific article, it is rather a project report because it has very few original results and these results are provided and discussed in a strange way. Please, find my comments and suggestions below.
Details
Abstract
It is immediately noticeable that there is hardly any original result in this study. Description of the materials and methods is also missing.
Line 17: ..
Introduction
The questions and hypotheses are missing. Later I realized because there are no real analyses which cannot include questions. The authors refer to some questions in line 126 but they are not listed.
I also miss the references. There are many-many statements and sentences which do not have references. It is very strange in a scientific article.
Line 59: “according to […]” is a strange way to cite a study. At least the first author should be mentioned such as in lines 64, 106.
Line 61: what should these articles have focused on besides physical and chemical properties?
Line 71: soil-forest relationship? What is that? Do you mean soil-vegetation relationship in a forest habitat?
Lines 81-87: verbs are missing to understand your principles.
Line 108: 91F0???? Why are you using this code here?
Lines 124, 126: .. ,.
Materials & Methods
In the section of materials and methods, the authors should describe all the activities, by which the data were collected and analysed and the results were displayed. Most of these descriptions are missing.
It is not clear where the data are from, what kind of data base. The section should have started with this information. Forest Data Bank was named only in line 142, 15 lines later than the authors started to write about the data.
There was no adequate description about the analyses of Table 2-3, Figure 2, Figures 4-6.
Line 152, Figure 1: what is NDVI? FDB? GUS 2021?
Figure 1: This flowchart should be described better in the text.
Results
It is very strange that this section has two parts, but the Materials and Methods does not have these two parts. I think M&M should also have this division which would make more understandable the manuscript.
Line 167-168: this information is in the Table (and almost the only result of this manuscript).
Line 175-176: sentences are for Introduction or to Discussion section.
Line 176-177: it is also very strange that we can read about 91F0 and 91E0 here first time.
Figure 2: It is also very strange that we cannot read anything about Figure 2. Why did the authors take this figure into the manuscript? Is there anything interesting in this figure? It should be detailed.
Lines 183-193: For Discussion.
Table 2: what about the other habitat types? What are the other habitat types?
Line 190: considering or assuming?
Line 197: “is obvious”????? Show evidence!
Line 198: “elements of nature”?
Line 199: “main form”?
Lines 212-229: These are not the authors’ results.
Lines 236-239: For discussion.
Line 240: Why didn’t you mention in M&M section that they will have this example of river Warta?
Lines 243-247: Not the results of the authors.
Line 255: Why didn’t the authors mention in M&M section that they will have this example of fluctuating water level of rivers?
Line 264: Natural factors?
Lines 280-283: Why didn’t the authors mention in M&M section that they will have this example of forested fluvisols?
Discussion:
The section Discussion deals with much broader topic than the results reported in Results section with many side talks and missing literature references.
Conclusions:
We can read here new information (such as lines 491-493) which is strange for a Conclusion section.
Lines 498, 502: how did you determine these statements?
Line 508: no data is available about this statement in this manuscript.
Line 511-517: this is the only result of this manuscript.
Line 518-531: Are these your results?
Comments on the Quality of English LanguageThe English grammar is good, however, there are many too long and hardly understandable sentences.
Author Response
Reviewer 4.
We would like to thank a lot the Reviewer for the review. Below are our responses to the Reviewer's comments given in the second round of revision.
Rev. 4. Abstract
It is immediately noticeable that there is hardly any original result in this study. Description of the materials and methods is also missing.
Authors' response: The missing data have been completed.
Rev. 4. Line 17: ..
Authors' response: The double dot has been replaced with a single dot.
Rev. 4. Introduction
The questions and hypotheses are missing. Later I realized because there are no real analyses which cannot include questions. The authors refer to some questions in line 126 but they are not listed.
Authors' response: The aim of the study was completed by the questions given in the Introduction.
Rev. 4. I also miss the references. There are many-many statements and sentences which do not have references. It is very strange in a scientific article.
Authors' response: The introduction has 98 lines and contains 31 references, that is, about 3 lines per citation. We think this is not a bad proportion.
Rev. 4. Line 59: “according to […]” is a strange way to cite a study. At least the first author should be mentioned such as in lines 64, 106.
Authors' response: Authors' names have been added to item [17] (line 61, earlier 59). In other places indicated by the Reviewer (lines 66, earlier 64; 114, earlier 106), the authors' names were included in the text (line 66 - Schürings et al. [3]; line 114 - Pechanec et al. [26]).
Rev. 4. Line 61: what should these articles have focused on besides physical and chemical properties?
Authors' response: The context of the sentence that “not for protection of these soils” has been added.
Rev. 4. Line 71: soil-forest relationship? What is that? Do you mean soil-vegetation relationship in a forest habitat?
Authors' response: A broad soil-forest relationship context was adopted, as for example in: “Protective functions of the forest - Relationship to soil and water conservation” (https://www.fao.org/3/x5372e/x5372e06.htm); Some Relationships Between Soil Type and Forest Site Quality (https://doi.org/10.2307/1932594; Mishra, et al. (2014). Relation of forest structure and soil properties in natural rehabilitated and degraded forest. Journal of Biodiversity Management & Forestry. 2.)
Rev. 4. Lines 81-87: verbs are missing to understand your principles.
Authors' response: The manuscript, after revision will be send to English proofreading.
Rev. 4. Line 108: 91F0???? Why are you using this code here?
Authors' response: We used the code because it is strictly defined as forests with specific natural value. To make it more understandable, we cited the data source [8].
Rev. 4. Lines 124, 126: .. ,.
Authors' response: The double dot has been replaced with a single dot.
Rev. 4. Materials & Methods
In the section of materials and methods, the authors should describe all the activities, by which the data were collected and analysed and the results were displayed. Most of these descriptions are missing.
It is not clear where the data are from, what kind of data base. The section should have started with this information. Forest Data Bank was named only in line 142, 15 lines later than the authors started to write about the data.
Authors' response: According to Reviewer's right comment, information on the source of data was put at the beginning of the section.
Rev. 4. There was no adequate description about the analyses of Table 2-3, Figure 2, Figures 4-6.
Authors' response: The description about the analyses of Table 2-3, Figure 2, Figures 4-6 is explained on Fig. 1 (Flowchart ..). E.g., Table 2 (actually 3) contains the data on forest habitat types. Figure 1 shows that the data comes from the FDB. However, taking into account the Reviewer's comments, a broader description of the database was added in the Methods chapter and we hope that now the data analysis methods will be more understandable.
Rev. 4. Line 152, Figure 1: what is NDVI? FDB? GUS 2021?
Authors' response: An explanation was added.
Rev. 4. Figure 1: This flowchart should be described better in the text.
Authors' response: The "methods" section has been reorganized and expanded.
Rev. 4. Results
It is very strange that this section has two parts, but the Materials and Methods does not have these two parts. I think M&M should also have this division which would make more understandable the manuscript.
Authors' response: According to the Reviewer's suggestion, M&M has been divided into two sections
Rev. 4. Line 167-168: this information is in the Table (and almost the only result of this manuscript).
Authors' response: The share and distribution of forested fluvisols, as well as their conservation status and environmental value, have not yet been described in the literature, so we believe that the results of our research are both the data in tables 2, 3 and those presented in figures 2, 4-6.
Rev. 4. Line 175-176: sentences are for Introduction or to Discussion section.
Authors' response: In our opinion, this sentence refers to the results of our research, so we leave it in the Results section.
Rev. 4. Line 176-177: it is also very strange that we can read about 91F0 and 91E0 here first time.
Authors' response: It is mentioned here for the first time, because it relates to the result of our research.
Rev. 4. Figure 2: It is also very strange that we cannot read anything about Figure 2. Why did the authors take this figure into the manuscript? Is there anything interesting in this figure? It should be detailed.
Authors' response: the aim of the study was to assess the state of knowledge about soils and forest ecosystems of river valleys in terms of the possibility of protecting the river valleys environment. Fig. 2 (actually 3) presents the share of forested fluvisols in various forms of protection against the background of their management methods, so it is related to the aim of the study.
Rev. 4. Lines 183-193: For Discussion.
Authors' response: It is the comment to Fig. 2 (actually 3). Therefore, if we leave Fig. 3 in the results, the text indicated by the Reviewer will also be in the results.
Rev. 4. Table 2: what about the other habitat types? What are the other habitat types?
Authors' response: The “other” has been explained in the table [Other (non Natura 2000) habitat types]
Rev. 4. Line 190: considering or assuming?
Authors' response: Reviewer is right. Assuming is better.
Rev. 4. Line 197: “is obvious”????? Show evidence!
Authors' response: Authors are surprised by the question. Is it not obvious that the role of national parks and nature reserves is to protect the best-preserved elements of nature, ensure the natural course of processes taking place in nature, and that forest management should be excluded?
The content of the Nature Protection Act may be evidence of this:
ACT of 16 April 2004 o Nature conservation (In Polish: https://isap.sejm.gov.pl/isap.nsf/DocDetails.xsp?id=wdu20040920880;
In English: https://www.global-regulation.com/translation/poland/3353949/the-act-of-16-april-2004-on-the-protection-of-nature.html
Chapter 1
General provisions
Article 1. The law lays down the objectives, principles and forms of the protection of the living and inanimate nature and the landscape.
Article 2.1. The protection of nature, within the meaning of the Act, consists in the preservation, sustainable use and renewal of resources, creations and natural components:
1) wildlife of plants, animals and fungi;
2) plants, animals and mushrooms covered by species protection;
……..
- The aim of nature conservation is:
1) maintaining ecological processes and the stability of ecosystems;
2) conservation of biodiversity;
3) conservation of geological and paleontological heritage;
4) ensuring the continuity of the existence of plant species, animals and fungi, together with their habitats, by their maintenance or restoration to the proper state of protection;
5) protection of landscape valors, greenery in towns and villages, and revelations;
6) maintaining or restoring to a proper state of conservation of natural habitats, as well as other resources, creations and nature components;
7) shaping the right attitude of man to nature through education, information and promotion in the field of nature conservation.
Rev. 4. Line 198: “elements of nature”?
Authors' response: Maybe “components” will be better. The manuscript will be checked by English Language Editing Services
Rev. 4. Line 199: “main form”?
Authors' response: “There are many different forms of protection for nature in Sweden. The most common are nature reserves and the strongest forms are the national park and Natura 2000” (https://www.naturvardsverket.se/en/topics/protected-areas/different-types-of-nature-conservation/)
Rev. 4. Lines 212-229: These are not the authors’ results.
Authors' response: Reviewer is right. These are not our results and we cited the source of citations (Smith and Smith [38]). We have quoted this text to justify Fig. 4
Rev. 4. Lines 236-239: For discussion.
Authors' response: We have decided that this text is necessary to justify Figures 5 and 6.
Rev. 4. Line 240: Why didn’t you mention in M&M section that they will have this example of river Warta?
Authors' response: We have updated this information in M&M.
Rev. 4. Lines 243-247: Not the results of the authors.
Authors' response: Yes, it is not our result, but it is public information necessary to description the Fig. 4.
Rev. 4. Line 255: Why didn’t the authors mention in M&M section that they will have this example of fluctuating water level of rivers?
Authors' response: The following description was added to the M&M:
The data was retrieved from https://danepubliczne.imgw.pl/data/dane_pomiarowo_obserwacyjne/dane_hydrologiczne/dobowe/. Daily data was used, but the folder structure did not allow for automatic utilization. The daily data were loaded into RStudio, and the Nietków station was filtered out. In the next step, the data was merged into a single series and loaded into Microsoft Excel for visualization.
Rev. 4. Line 264: Natural factors?
Authors' response: Man activity may overlap with natural factors like e.g. droughts
Rev. 4. Lines 280-283: Why didn’t the authors mention in M&M section that they will have this example of forested fluvisols?
Authors' response: The description of M&M has been expanded. In section 2.1, the authors refer to the distribution of fluvisols. The result of our investigation is Fig. 7.
Rev. 4. Discussion:
The section Discussion deals with much broader topic than the results reported in Results section with many side talks and missing literature references.
Authors' response: We respect the Reviewer's opinion, but we think that since the aim of the study is to "assess the state of knowledge about soils and forest ecosystems of river valleys in terms of the possibility of protecting the river valleys environment", the broader context of the discussion is necessary for the answer for the purpose of research. Besides this, we do not know what references the Reviewer considers omitted, because all data sources have been cited.
Rev. 4. Conclusions:
We can read here new information (such as lines 491-493) which is strange for a Conclusion section.
Authors' response: The information was added to the M&M.
Rev. 4. Lines 498, 502: how did you determine these statements?
Authors' response: According to line 498 (“Riparian forests have high natural value, protecting both biodiversity and natural processes occurring in the associated soils”) this is due to the high share of habitats of high nature value (91F0, 91E0 - Table 3).
According to line 502 (“the natural value of riparian forests may significantly exceed their economic value”) this is due to the comparison of Table 4 and a calculation based on Pechanec et al [47] paper. That was described in the Discussion section, in the Environmental and monetary value of alluvial forest habitats subsection.
Rev. 4. Line 508: no data is available about this statement in this manuscript.
Authors' response: The conclusion was modified.
Rev. 4. Line 511-517: this is the only result of this manuscript.
Authors' response: This is the main result from which the remaining conclusions are drawn.
Rev. 4. Line 518-531: Are these your results?
Authors' response: No, it is our conclusion
Round 2
Reviewer 2 Report (Previous Reviewer 2)
Comments and Suggestions for Authors
All the comments have been answered or clarified by the authors. The manuscript can be published
Author Response
Thank you very much!
This manuscript is a resubmission of an earlier submission. The following is a list of the peer review reports and author responses from that submission.
Round 1
Reviewer 1 Report
Comments and Suggestions for Authors
This study reported an interesting topic about the relationship between forest and fluvisols in Poland. Generally, the paper is well written and the results have practical values. Here are some suggestions that may be helpful for the improvement of the manuscript.
1. The title. “The role of forests in protecting the soils of river valleys”. This study is mainly about the distribution of forest and its relationship with fluvisols. It seems that no specific soil properties data have been used to show how forests would protect the soil. I suggest changing the title to make sure it fits the main theme of this study.
2. Abstract. L18-22, these contents should be strengthened to ensure that most of the important results are included.
3. The structure of the introduction section should be improved. I suggest moving the content from L59 to L61 to the end of the introduction section. Furthermore, the objectives and novelty of this study should be explained based on a detailed literature review.
4. The material and method section
More information about the spatial analysis methods should be provided.
5. The results section.
Subtitles should be used to show the readers the main structure of the results section.
6. The discussion section
The relationship between the results and the discussion, as well as between the contents of discussion in the different section, needs to be strengthened. For example, what is the connection between “Environmental value of alluvial forest habitats.”, “Selection of regions and goals of nature protection in river basins.” and “Management of the river valleys”?
7. The conclusion
More quantitative results should be used to support the conclusions in the third paragraph (L372-381).
8. Other minor issues
Superscripts should be used in the case of area units.
L274, maybe the area of the Yangtze River Basin is too large.
Reviewer 2 Report
Comments and Suggestions for Authors
Brief summary
This manuscript describes the share and distribution of Fluvisols in Poland. This manuscript is suitable for publication in “Land”. However, there are some aspects of the manuscript that should be improved before considering it for publication. Its main flaw is the lack of originality and the local importance of the study (Poland).
Broad comments
The title is OK.
Keywords: 91F0 is too specific.
The introduction should be written in a clear way. The knowledge gap is the lack of study of distribution of Fluvisols in Poland. This gives it a local perspective. Authors should make a better effort to pinpoint the limitations or contradictions of previous studies to show the knowledge gap, and therefore how their study will generate original knowledge. The aim of the manuscript is to study the share and distribution of Luvisols in Poland using databases and GIS; however, it seems more appropriate for a report rather than a research manuscript.
Some specific comments
L12. Not in world literature, only in the WRB.
L28. What is Natura 2000?
L87. In what sense are the opinions divided?
L96-97. It is not necessary to show the extension of the files and the license agreement.
L122. Table 1 should be deleted. This information can be discussed in the main text.
L126. One-sentence paragraph.
Reviewer 3 Report
Comments and Suggestions for Authors
Here is the review report for manuscript entitled “The role of forests in protecting the soils of river valleys”. Overall, the manuscript presents interesting insights into the protecting the soils. However, further clarification and expansion of certain aspects would greatly enhance the quality and impact of the study. The below questions and comments should be considered and addressed by authors before the next stage.
1- Is there existing literature on the role of forests in protecting the soils of river valleys? How does this paper contribute to the current knowledge? The authors should provide a comprehensive review of existing literature to establish the significance of their study and clearly state how their research fills the gaps in current knowledge.
2- How were the forested fluvisols (FLs) in Poland identified and classified? Were there any specific criteria or parameters used for their recognition? The authors should provide detailed information on the methodology used for identifying and classifying forested fluvisols in Poland to ensure the accuracy and reliability of their findings.
3- In the results section, it is mentioned that most FLs are located in Natura 2000 sites. Can the authors explain the reasons behind this distribution pattern? Are there any specific ecological or environmental factors influencing this distribution?
4- The paper mentions that the effectiveness of FLs' conservation was limited by the lack of integrated conservation plans covering the entire river basin. Can the authors elaborate on the potential challenges and barriers to implementing such integrated conservation plans? The authors should provide a detailed discussion on the challenges and barriers faced in implementing integrated conservation plans for FLs in river basins, including any legal, administrative, or institutional limitations that hinder effective conservation strategies.
5- The abstract mentions that the study assessed the relationship of FLs to Natura 2000 forest habitats. Can you provide more information on this relationship? How do FLs contribute to the conservation and management of Natura 2000 forest habitats?
6- The paper discusses the importance of riparian forests (RFs) in the water cycle and soil protection. Present specific examples or case studies that demonstrate the influence of RFs on soil protection.
7- What data sources and sampling methods were used to obtain the 17,820 records from the Polish Forest Data Bank? Are there any limitations or biases associated with the data collection process that should be addressed? Provide a detailed explanation of the data sources and sampling methods used to collect the records from the Polish Forest Data Bank. Additionally, discuss any potential limitations or biases in the data that may affect the accuracy or generalizability of their findings.
8- The paper mentions that FLs often remain in agricultural use, which changes their structure and properties. Can you provide more information on the specific impacts of agricultural use on FLs? How do these changes affect soil fertility and conservation? You may present a comprehensive analysis of the effects of agricultural use on FLs, including the specific changes in soil structure and properties and their implications for soil fertility and conservation. This analysis will help readers understand the potential consequences of agricultural practices on riparian soil ecosystems.
9- The abstract mentions the assessment of the state of protection of forested FLs. Can you elaborate on the specific criteria or indicators used to evaluate the state of protection? How was the effectiveness of protection measured?
10- The paper discusses the importance of forested FLs as the basis for the management strategy of river valleys. Can you provide practical recommendations or guidelines for the effective management and conservation of FLs in river valleys? You may consider providing practical recommendations or guidelines for the management and conservation of forested FLs in river valleys based on their study findings. This will enhance the applicability and relevance of the research and provide insights for policymakers and practitioners.
11- The paper mentions the lack of studies on floodplain protection and a general lack of studies at the regional scale on FLs. Can you discuss the potential reasons behind this research gap and the implications for future research and conservation efforts?
12- The paper states that FLs are classified as fluvisols. Can you provide a brief explanation of the key characteristics and properties of fluvisols? How do these characteristics contribute to their role in protecting river valley soils?
13- The abstract mentions the aim of the study was to recognize the share and distribution of forested FLs in Poland. Can you explain the significance of understanding the share and distribution of FLs? How can this information contribute to the management and conservation of river valleys? You may should elaborate on the practical implications of recognizing the share and distribution of forested FLs in Poland. Also, you should discuss how this information can be utilized to inform management and conservation strategies for river valleys, including the identification of priority areas for protection or restoration.
14- The paper highlights the inclusion of riparian forests in the European Natura 2000 network. Can you provide more details on the specific objectives and goals of Natura 2000 with respect to riparian forests? How does the conservation of FLs align with these objectives?
15- The paper mentions the importance of integrated conservation plans for effective protection of FLs. Present examples or case studies where integrated conservation plans have been successfully implemented in river basins. What lessons can be learned from these examples?
16- The abstract mentions the assessment of the relationship between FLs and Natura 2000 forest habitats. Elaborate on the ecological connections between FLs and these forest habitats. How do FLs support the biodiversity and ecological functions of Natura 2000 forest habitats?
Comments on the Quality of English LanguageFair enough
Reviewer 4 Report
Comments and Suggestions for Authors
This paper looks at the distribution and protection of forested fluvisols in Poland, highlighting their occurrence in Natura 2000 sites, and the challenges in conserving these ecosystems. Integrated management strategies in river valleys are emphasized, while taking into account factors such as agriculture and hydrotechnical structures in these habitats.
The Introduction is reasonably well written. However, I think the transition from introducing the importance of riparian forests to alluvial soils needs to be smoother. Please consider improving the flow.
Materials and Methods doesn't look complete and needs significant improvements. There isn't nearly enough information so that I can reproduce this research. A detailed methodology would help with reproducibility, but also lend credibility to the study. Please follow the style for website references. Listing file type extensions should use commas to separate. Only two references doesn't seem nearly enough. More transparency is needed with regards to how data was collected or analysed.
More discussion on the significance of the findings in the broader context of forest conservation would improve the section discussion section. The study focuses on Poland, are the findings generalizable to other regions with similar ecological characteristics?
The comparison of FLs and RFs coverage in Europe and Poland, and their environmental significance, provides a good insight into the broader relevance of the study. However, more direct comparisons with specific studies or data from other regions would help improve the article.
Mentioning the need for urgent research and an integrated management plan for riparian forests in Poland is an important point, however, I think the section needs to be more specific with suggestions or directions for future research.
My main concern regarding the paper is that it's very descriptive without much quantitative analysis of the data. I think more in-depth statistical analyses would provide deeper insights into the relationships between forested fluvisols and various environmental factors, or the effectiveness of different conservation strategies. Including more quantitative analyses would help to draw more robust and generalizable conclusions, therefore strengthening the study's overall impact.
Reviewer 5 Report
Comments and Suggestions for Authors
RFs are most valuable as well as vulnerable eco-system. Integrated protection plan definitely has high priority. The issue discussed in the paper is definitely country specific. In flood prone river valleys, where bank erosion may be a critical issue the protection plan will be quite different. It is definitely a good discussion in the limited context but needs wider scientific analyses to address universal issues of the vulnerability of FL soils and RFs.